

# Nomogram for predicting cancer specific survival in inflammatory breast carcinoma: a SEER population-based study

Haige Zhang[1,*], Guifen Ma[2,*], Shisuo Du[2], Jing Sun[2], Qian Zhang[2], Baoying Yuan[2] and Xiaoyong Luo[1]

[1] Department of Radiation Oncology, Luoyang Central Hospital affiliated to Zhengzhou University, Luoyang, China
[2] Department of Radiation Oncology, Zhongshan Hospital, Fudan University, Shanghai, China
[*] These authors contributed equally to this work.

## ABSTRACT

The clinicopathological features of inflammatory breast carcinoma (IBC), the effect of therapeutic options on survival outcome and the identification of prognostic factors were investigated in this study. Information on IBC patients were extracted from the Surveillance, Epidemiology, and End Results (SEER) database between 2010 and 2015. Cox proportional hazard regression was used to determine potential significant prognostic factors of IBC. A nomogram was then constructed to evaluate patient survival based on certain variables. Univariate and multivariate analyses revealed that race ($p < 0.001$), M stage ($p < 0.001$), surgery ($p = 0.010$), chemotherapy (CT) ($p < 0.001$), tumor size ($p = 0.010$), estrogen receptor ($p < 0.001$), progesterone receptor ($p = 0.04$), and human epidermal growth factor receptor 2 ($p < 0.001$) were all independent risk factors. The concordance index (C-index) of the nomogram was 0.735, which showed good predictive efficiency. Survival analysis indicated that IBC patients without CT had poorer survival than those with CT ($p < 0.001$). Stratified analyses showed that modified radical mastectomy (MRM) had significant survival advantages over non-MRM in patients with stage IV IBC ($p = 0.031$). Patients treated with or without CT stratified by stage III and stage IV showed better survival than those without stage III and IV ($p < 0.001$). Trimodality therapy resulted in better survival than surgery combined with CT or CT alone ($p < 0.001$). Competing risk analysis also showed the same results. The nomogram was effectively applied to predict the 1, 3 and 5-year survival of IBC. Our nomogram showed relatively good accuracy with a C-index of 0.735 and is a visualized individually predictive tool for prognosis. Treatment strategy greatly affected the survival of patients. Trimodality therapy was the preferable therapeutic strategy for IBC. Further prospective studies are needed to validate these findings.

## INTRODUCTION

Inflammatory breast carcinoma (IBC) is an uncommon subtype of locally advanced breast cancer with a poor prognosis and is characterized by aggressive behavior and rapid progression (*Robertson et al., 2010*). Management involves the coordination of multidisciplinary treatment and usually includes neoadjuvant chemotherapy, ablative surgery if a tumor-free resection margin is expected and locoregional radiotherapy. This multimodal therapeutic approach has significantly improved patient survival. However, the median overall survival (OS) among women with IBC is still poor. Clinically, IBC is characterized by the rapid onset of breast warmth, erythema, and edema often without a well-defined mass. Along with extensive breast involvement, women with IBC often have early involvement of the axillary lymph nodes. The median survival time of IBC treated by surgery alone or combined with radiotherapy (RT) is less than 15 months and the local recurrence rate is approximately 50% (*Zucali et al., 1976*). Despite the significant progress made in the treatment of this aggressive form of breast cancer, most women with IBC will relapse and succumb to this disease. Limited research with small cohorts have focused on the prognostic value of RT and CT (*Semiglazov et al., 2007*; *Jardel et al., 2018*). A few studies have analyzed the clinicopathological features related to OS, such as lymph node status and human epidermal growth factor receptor 2 (Her-2) status (*Dawood et al., 2008*; *Wecsler et al., 2015*). It is important to evaluate the effect of trimodality therapy in IBC patients. In addition, high risk patients who need more aggressive locoregional treatment are not well-defined in the literature. The current investigation aimed to determine the clinicopathological features of IBC and identify reliable and comprehensive prognostic factors based on a large cohort, which will help to predict the survival outcome of patients with IBC and choose the appropriate treatment strategy.

Previous studies of IBC patients from the SEER database have reported that breast cancer subtype is clinically useful for predicting survival outcome in IBC (*Wu et al., 2019*). The identifying factor of survival was related to race/ethnic group (*Il'Yasova et al., 2011*) and the overall survival IBC patients from the SEER database has been reported to be associated with socioeconomic position (*Liu, Deapen & Bernstein, 1998*; *Schlichting et al., 2012*). In contrast, we determined the independent prognostic factors of IBC, and then constructed a nomogram.

A nomogram, a statistic-based tool to calculate the risk of clinicopathological features of a cancer, has been widely developed to evaluate survival in cancer patients (*Zhou et al., 2015*; *Sun et al., 2017*). It has been proved to be accurate with the advantage of visualization and quantification with a friendly interface for doctors and patients. In this study, a nomogram was constructed to predict 1-, 3-, and 5-year cancer-specific survival (CSS) according to multivariate analyses based on this large cohort, and to identify the exact therapeutic effect of various surgical procedures, RT and CT.

## MATERIALS AND METHODS

### Data source and inclusion criteria

We retrieved patient data from the SEER database (SEER*Stat 8.3.5, http://seer.cancer.gov/). The SEER database is free to the public and is updated annually, with routinely collected general messages from patients, primary tumor characteristics, treatments, survival and follow-up, etc. In this study, the data were updated in November 2016, and released on April 16, 2018. The target population downloaded from the database was from 1973 to 2015. We extracted IBC patients between 2010 and 2015 as data on Her-2 status was released from 2010. Patients were included in this study if they met the following criteria: patients who had IBC with the International Classification of Diseases for Oncology, 3rd edition (ICD-O-3): 8530 as the histopathology code, positive histology, complete survival month flag and active follow-up. Patients confirmed by autopsy only or death certificate only and those with invalid follow-up data were excluded. In total, 883 patients were selected for further analysis in this study. Data downloaded from SEER did not require patients' informed consent and may be reproduced or copied without permission.

### Endpoints and statistical analysis

OS was calculated from the date of first diagnosis to the date of death or last contact. CSS was measured from the date of first diagnosis to the date of death due to breast carcinoma. However, patients who died of other causes were considered censoring events. All statistical analyses were performed using SPSS 24.0 (IBM Corp, Armonk, NY, USA) or R package 3.4.4 software (*R Core Team, 2018*). Cox proportional hazard regression was applied to identify significant prognostic factors with hazard ratios (HRs) and 95% confidence intervals (95% CIs). Variables in the univariate analysis with $p$-values $< 0.05$ were selected for multivariate analysis using backward stepwise regression (likelihood ratio). The nomogram was constructed based on multivariate analysis results, and was evaluated by the concordance index (C-index), the calibration curves and receiver operating characteristics (ROC) curves. The value of the C-index ranges from 0.5 to 1.0 and a larger value indicates better accuracy of predicting efficiency (*Koepsell & Weiss, 2014*; *Ma et al., 2018*). The calibration curves were based on 1,000 times bootstrap resampling. The Kaplan–Meier method and log-rank test were performed to obtain survival curves and the differences between groups were compared using R project 3.4.4 with the survival package. We also performed competing risk analysis. All differences with $p$ values $< 0.05$ were considered to be statistically significant.

## RESULTS

### Patient characteristics

A total of 883 patients diagnosed with IBC were extracted from the SEER database between 2010 and 2015 according to the abovementioned inclusion criteria. The results of univariate/multivariate analyses based on CSS and OS as well as 5-year OS are summarized in Tables 1–3, respectively. The median age of the IBC patients was 57 (range, 22–97) years. The majority (693, 78.5%) were White. The 5-year OS of White vs. Black patients was

49.4% vs. 29.7% and the HR for CSS was 1.90 (95% CI [1.43–2.53], $p < 0.001$). The 3- and 5-year OS of patients with IBC was 54.4% and 47.4%, respectively. The median OS was 43.0 (95% CI [32.4–53.6]) months. Tumor staging followed the criteria of the 7th edition of American Joint Committee on Cancer (AJCC) for breast cancer. Most patients (572, 64.8%) were stage III, while 311 (35.2%) were stage IV. Patients who were treated with surgery accounted for 58.2%, of which, 352 (39.9%) patients underwent MRM and 162 (18.3%) patients were treated by non-MRM including 30 (3.0%) with breast-conserving surgery, 118 (13.4%) with total (simple) mastectomy, 9 (1.0%) with radical mastectomy and 5 (0.9%) with extended radical mastectomy. The HR and 95% CI of the non-MRM group vs. the MRM group was 0.41 (0.30–0.57) vs. 0.33 (0.26–0.43) for CSS. RT (396/883, 44.8%) and CT (721/883, 81.7%) were also main treatments for patients with IBC.

Univariate analysis of CSS for IBC using Cox regression analysis demonstrated that age ($p = 0.016$), race ($p < 0.001$), N stage ($p < 0.001$), M stage ($p < 0.001$), surgical modality ($p < 0.001$), radiation status ($p < 0.001$), CT status ($p < 0.001$), tumor size ($p = 0.015$), estrogen receptor (ER) ($p < 0.001$), progesterone receptor (PR) ($p < 0.001$), Her-2 ($p < 0.001$) and marital status ($p = 0.002$) were all significant prognostic factors, and corresponded to those for OS. All the significantly different variables were included in the multivariate analyses of CSS and OS (Tables 1 and 2). Finally, race (Black, HR = 1.77), M stage (M1, HR = 2.93), surgery (non-MRM, HR = 0.74; MRM, HR = 0.60), CT status (Yes, HR = 0.46), tumor size (2–5 cm, HR = 0.91; >5 cm, HR = 1.48; Diffuse, HR = 1.39), ER (Positive, HR = 0.52), PR (Positive, HR = 0.64) and Her-2 (Positive, HR = 0.12) were confirmed as independent prognostic factors of IBC.

A nomogram was constructed to evaluate patient survival based on multivariate analysis (Fig. 1A). The points from each independent prognostic factor listed in the nomogram were added together. The 1, 3 and 5-year CSS rates for patients with IBC were estimated by applying three point scales at the bottom of the nomogram, respectively. The C-index of this model was 0.735 which showed an excellent predictive efficiency. We performed a calibration plot to resample the validation of the nomogram based on 1,000 bootstraps to confirm the agreement between the actual and predicted 1, 3 and 5-year CSS rates (Figs. 1B–1D). A series of receiver operating characteristics (ROC) curves were performed to verify the accuracy of the nomogram (Figs. S1A–S1C).

## Survival analyses in subgroups of patients with inflammatory breast carcinoma

Survival curves were used to compare the CSS of patients with IBC according to the following parameters: TNM stage, CT, surgical modality, and treatment pattern by subgroup analyses. All 883 patients classified by different surgical modalities achieved statistically significant survival ($p < 0.001$) (Fig. 2A). There was no significant difference in CSS between the MRM and non-MRM group ($p = 0.23$) (Fig. 2B). When the patients were stratified by TNM stage III and IV, the $p$ values were 0.5 (Fig. 2C) and 0.031 (Fig. 2D), respectively. Patients treated with CT had better survival ($p < 0.001$) (Fig. 3A). Competing risk analysis showed the same results (Fig. S2). Furthermore, subgroup analyses stratified by TNM stages (Figs. 3B and 3C) indicated that patients in the TNM stages benefitted

**Table 1  Patient characteristics and analyses based on cancer specific survival status.**

| Patient characteristics | Number (%) | Univariate analysis | | Multivariate analysis | |
|---|---|---|---|---|---|
| | | HR (95% CI) | *P*-value | HR (95% CI) | *P*-value |
| Age | | | 0.016 | | |
| <50 | 234 (26.5) | 1 [Reference] | | | |
| 50–59 | 262 (29.7) | 1.03 (0.76–1.40) | 0.857 | | |
| 60–69 | 199 (22.5) | 1.10 (0.79–1.52) | 0.583 | | |
| ≥70 | 188 (21.3) | 1.59 (1.15–2.18) | 0.005 | | |
| Race | | | <0.001 | | <0.001 |
| White | 693 (78.5) | 1 [Reference] | | 1 [Reference] | |
| Black | 123 (13.9) | 1.90 (1.43–2.53) | <0.001 | 1.77 (1.31–2.38) | <0.001 |
| Unknown | 67 (7.6) | 1.10 (0.71–1.68) | 0.680 | 1.09 (0.70–1.68) | 0.705 |
| N stage | | | <0.001 | | |
| N0 | 144 (16.3) | 1 [Reference] | | | |
| N1 | 372 (42.1) | 1.49 (1.02–2.18) | 0.038 | | |
| N2 | 138 (15.6) | 1.38 (0.89–2.16) | 0.152 | | |
| N3 | 177 (20.0) | 1.85 (1.22–2.78) | 0.003 | | |
| $N_X$ | 52 (5.9) | 3.08 (1.87–5.07) | <0.001 | | |
| M stage | | | <0.001 | | <0.001 |
| M0 | 572 (64.8) | 1 [Reference] | | 1 [Reference] | |
| M1 | 311 (35.2) | 3.23 (2.57–4.06) | <0.001 | 2.93 (2.27–3.79) | <0.001 |
| Surgery | | | <0.001 | | 0.010 |
| No surgery | 355 (40.2) | 1 [Reference] | | 1 [Reference] | |
| Not MRM | 162 (18.3) | 0.41 (0.30–0.57) | <0.001 | 0.74 (0.52–1.04) | 0.086 |
| MRM | 352 (39.9) | 0.33 (0.26–0.43) | <0.001 | 0.60 (0.45–0.82) | <0.001 |
| Unknown | 14 (1.6) | 0.96 (0.47–1.96) | 0.911 | 1.11 (0.54–2.30) | 0.778 |
| Radiation | | | <0.001 | | |
| No | 487 (55.2) | 1 [Reference] | | | |
| Yes | 396 (44.8) | 0.57 (0.45–0.72) | <0.001 | | |
| Chemotherapy | | | <0.001 | | <0.001 |
| No | 162 (18.3) | 1 [Reference] | | 1 [Reference] | |
| Yes | 721 (81.7) | 0.41 (0.32–0.54) | <0.001 | 0.46 (0.34–0.63) | <0.001 |
| Tumor size | | | 0.015 | | 0.010 |
| <2 cm | 68 (7.7) | 1 [Reference] | | 1 [Reference] | |
| 2–5 cm | 186 (21.1) | 0.90 (0.55–1.47) | 0.664 | 0.91 (0.55–1.50) | 0.715 |
| >5 cm | 257 (29.1) | 1.40 (0.89–2.20) | 0.151 | 1.48 (0.94–2.35) | 0.093 |
| Diffuse | 179 (20.3) | 1.61 (1.00–2.57) | 0.048 | 1.39 (0.86–2.25) | 0.178 |
| Unknown | 193 (21.9) | 1.21 (0.75–1.96) | 0.429 | 0.95 (0.57–1.57) | 0.834 |
| ER | | | <0.001 | | <0.001 |
| Negative | 386 (43.7) | 1 [Reference] | | 1 [Reference] | |
| Positive | 455 (51.5) | 0.54 (0.42–0.68) | <0.001 | 0.52 (0.37–0.71) | <0.001 |
| Unknown | 42 (4.8) | 1.30 (0.82–2.07) | 0.267 | 1.22 (0.37–4.06) | 0.746 |

**Table 1** (*continued*)

| Patient characteristics | Number (%) | Univariate analysis | | Multivariate analysis | |
|---|---|---|---|---|---|
| | | HR (95% CI) | *P*-value | HR (95% CI) | *P*-value |
| PR | | | <0.001 | | 0.04 |
| Negative | 499 (56.5) | 1 [Reference] | | 1 [Reference] | |
| Positive | 331 (37.5) | 0.60 (0.46–0.77) | <0.001 | 0.64 (0.45–0.91) | 0.013 |
| Other* | 53 (6.0) | 1.28 (0.84–1.95) | 0.261 | 0.61 (0.22–1.69) | 0.337 |
| Her-2 | | | <0.001 | | <0.001 |
| Negative | 496 (56.2) | 1 [Reference] | | 1 [Reference] | |
| Positive | 311 (35.2) | 0.49 (0.38–0.65) | <0.001 | 0.12 (0.04–0.39) | <0.001 |
| Other* | 76 (8.6) | 1.26 (0.87–1.83) | 0.221 | 1.16 (0.66–2.03) | 0.617 |
| Marital status | | | 0.002 | | |
| Married | 400 (45.3) | 1 [Reference] | | | |
| Divorce or widow | 238 (27) | 1.29 (0.97–1.71) | 0.078 | | |
| Single | 201 (22.8) | 1.47 (1.10–1.95) | 0.008 | | |
| Unknown | 44 (5.0) | 2.10 (1.35–3.29) | <0.001 | | |

**Notes.**

Abbreviations: ER, estrogen receptor; PR, progesterone receptor; HER2, human epidermal growth factor receptor 2; MRM, Modified radical mastectomy; *Other, borderline + unknown; Reference, this group is set as a reference group.

from CT in terms of CSS ($p < 0.001$). When patients were classified by different treatment patterns (Fig. 4), such as surgery alone, RT alone, CT alone, surgery combined with RT, surgery combined with CT, RT combined with CT, or surgery combined with RT and CT, there were significant differences between them. The results indicated that surgery combined with RT and CT was the optimal treatment pattern. A further subgroup analysis of the treatment modalities was performed and the treatments which resulted in greatest survival (Fig. 4) were surgery combined with RT and CT, followed by CT and RT, and CT alone with significant differences between the three groups ($p < 0.001$). Competing analysis revealed the same results (Fig. S3).

## DISCUSSION

IBC is uncommon and the most lethal form of breast carcinoma, and leads to a worse prognosis compared with other forms of breast carcinoma, with significantly lower OS, (3-year OS, 42% vs. 85%) (*Chang et al., 1998*). Treatment with multimodal therapies has significantly improved the survival of patients with IBC in recent years, especially when targeted therapy is available (*Gianni et al., 2010*). However, survival is still poor with a 5-year OS of 30–40% (*Hance et al., 2005*; *Ueno et al., 1997*). The 3- and 5-year OS of patients with IBC was 54.4% and 47.4% in this study, which is consistent with the above previously published reports. To date, there are still many uncertainties regarding the optimal treatment of IBC. As a result of small sample sizes, various treatment patterns, and variable response criteria, evidence-based management has been determined largely by institutional experience or based on other types of breast carcinoma (*Panades et al., 2005*). This study analyzed the survival of a large cohort of patients with IBC based on the SEER database and confirmed eight independent prognostic factors including race, M stage, surgical modality, CT status, tumor size, ER, PR and Her-2 status, based on the nomogram

**Table 2  Patient characteristics and analyses based on overall survival status.**

| Patient characteristics | Number (%) | Univariate analysis | | Multivariate analysis | |
|---|---|---|---|---|---|
| | | HR (95% CI) | *P*-value | HR (95% CI) | *P*-value |
| Age | | | <0.001 | | |
| <50 | 234 (26.5) | 1 [Reference] | | | |
| 50–59 | 262 (29.7) | 1.06 (0.78–1.43) | 0.727 | | |
| 60–69 | 199 (22.5) | 1.21 (0.88–1.65) | 0.237 | | |
| ≥70 | 188 (21.3) | 1.91 (1.42–2.58) | <0.001 | | |
| Race | | | <0.001 | | 0.003 |
| White | 693 (78.5) | 1 [Reference] | | 1 [Reference] | |
| Black | 123 (13.9) | 1.70 (1.29–2.24) | <0.001 | 1.64 (1.23–2.18) | 0.001 |
| Unknown | 67 (7.6) | 0.98 (0.65–1.49) | 0.093 | 0.99 (0.65–1.52) | 0.993 |
| N stage | | | <0.001 | | |
| N0 | 144 (16.3) | 1 [Reference] | | | |
| N1 | 372 (42.1) | 1.35 (0.95–1.91) | 0.096 | | |
| N2 | 138 (15.6) | 1.34 (0.90–2.02) | 0.153 | | |
| N3 | 177 (20.0) | 1.64 (1.12–2.40) | 0.011 | | |
| $N_X$ | 52 (5.9) | 3.21 (2.04–5.05) | <0.001 | | |
| M stage | | | <0.001 | | <0.001 |
| M0 | 572 (64.8) | 1 [Reference] | | 1 [Reference] | |
| M1 | 311 (35.2) | 2.98 (2.40–3.69) | <0.001 | 2.58 (2.03–3.29) | <0.001 |
| Surgery | | | <0.001 | | 0.004 |
| No surgery | 355 (40.2) | 1 [Reference] | | 1 [Reference] | |
| Not MRM | 162 (18.3) | 0.39 (0.29–0.54) | <0.001 | 0.71 (0.51–0.99) | 0.043 |
| MRM | 352 (39.9) | 0.33 (0.26–0.42) | <0.001 | 0.59 (0.45–0.79) | <0.001 |
| Unknown | 14 (1.6) | 0.84 (0.41–1.71) | 0.633 | 0.94 (0.45–1.94) | 0.864 |
| Radiation | | | <0.001 | | |
| No | 487 (55.2) | 1 [Reference] | | | |
| Yes | 396 (44.8) | 0.57 (0.46–0.71) | <0.001 | | |
| Chemotherapy | | | <0.001 | | <0.001 |
| No | 162 (18.3) | 1 [Reference] | | 1 [Reference] | |
| Yes | 721 (81.7) | 0.35 (0.28–0.44) | <0.001 | 0.39 (0.29–0.51) | <0.001 |
| Tumor size | | | 0.025 | | 0.033 |
| <2 cm | 68 (7.7) | 1 [Reference] | | 1 [Reference] | |
| 2–5 cm | 186 (21.1) | 0.97 (0.61–1.55) | 0.904 | 0.97 (0.61–1.56) | 0.907 |
| >5 cm | 257 (29.1) | 1.39 (0.90–2.15) | 0.140 | 1.47 (0.95–2.29) | 0.870 |
| Diffuse | 179 (20.3) | 1.63 (1.04–2.55) | 0.035 | 1.43 (0.90–2.56) | 0.131 |
| Unknown | 193 (21.9) | 1.33 (0.85–2.09) | 0.218 | 1.07 (0.66–1.72) | 0.786 |
| ER | | | <0.001 | | |
| Negative | 386 (43.7) | 1 [Reference] | | 1 [Reference] | |
| Positive | 455 (51.5) | 0.59 (0.47–0.73) | <0.001 | 0.56 (0.41–0.76) | <0.001 |
| Unknown | 42 (4.8) | 1.33 (0.85–2.07) | 0.211 | 1.02 (0.34–3.05) | 0.972 |

**Table 2** (*continued*)

| Patient characteristics | Number (%) | Univariate analysis | | Multivariate analysis | |
|---|---|---|---|---|---|
| | | HR (95% CI) | *P*-value | HR (95% CI) | *P*-value |
| PR | | | <0.001 | | 0.021 |
|   Negative | 499 (56.5) | 1 [Reference] | | 1 [Reference] | |
|   Positive | 331 (37.5) | 0.64 (0.51–0.82) | <0.001 | 0.63 (0.45–0.87) | 0.006 |
|   Other* | 53 (6.0) | 1.31 (0.88–1.95) | 0.19 | 0.70 (0.28–1.75) | 0.445 |
| Her-2 | | | <0.001 | | <0.001 |
|   Negative | 496 (56.2) | 1 [Reference] | | 1 [Reference] | |
|   Positive | 311 (35.2) | 0.51 (0.39–0.65) | <0.001 | 0.46 (0.35–0.60) | <0.001 |
|   Other* | 76 (8.6) | 1.27 (0.89–1.81) | 0.18 | 1.12 (0.66–1.88) | 0.677 |
| Marital status | | | 0.002 | | |
|   Married | 400 (45.3) | 1 [Reference] | | | |
|   Divorce or widow | 238 (27) | 1.47 (1.14–1.90) | 0.003 | | |
|   Single | 201 (22.8) | 1.40 (1.06–1.84) | 0.017 | | |
|   Unknown | 44 (5.0) | 1.91 (1.22–2.97) | 0.004 | | |

**Notes.**

Other*, borderline + unknown.

which will provide the first comprehensive evaluation profile to help physicians make a reasonable treatment decision and estimate prognosis in IBC patients.

ER, PR and Her-2 status are important prognostic factors in IBC. Endocrine therapy is an important treatment strategy for patients with breast carcinoma who are either ER or PR positive. However, information on endocrine therapy in IBC patients is not available in the SEER database. Her-2, is an oncogene and is overexpressed and/or amplified in approximately 30% of patients with breast carcinoma (*Slamon et al., 1987*), and is related to increased aggressiveness, a higher recurrence rate and mortality (*Romond, 2005*). Furthermore, it was reported that Her-2 was overexpressed and amplified by 36–60% in patients with IBC than in those with non-IBC (*Parton et al., 2004*; *Guerin et al., 1990*). Patients with positive expression of Her-2 showed shorter recurrence-free survival and OS compared with patients without Her-2 amplification in a study of breast carcinoma by *Slamon et al. (1987)*. However, the role of Her-2 as a poor prognostic factor in breast carcinoma was altered by the introduction of trastuzumab, a monoclonal antibody which targets the Her-2 receptor (*Wecsler et al., 2015*). Data from the SEER database showed that IBC patients with HoR+/Her2- subtype had poorer breast cancer-specific survival and OS than those with HoR+/Her-2 subtype (*Wu et al., 2019*). Her2-positive status in this study had a protective effect in patients with IBC, which was in accordance with previous reports based on the SEER database (*Li et al., 2017*).

Surgery and RT are important locoregional therapies in IBC. However, either surgery alone (*Fields et al., 1989*) or RT alone (*Jaiyesimi, Buzdar & Hortobagyi, 1992*) produced disappointing results in the treatment of IBC. Even though combining surgery with RT significantly improves locoregional control, with disease-free survival as high as 24 months (*Perez & Fields, 1987*), the median survival of the two treatments combined ranges from 7 to 29 months and was not significantly different from either treatment alone. Data on IBC patients from the SEER database showed that surgery combined with RT

**Table 3   List of 5-year overall survival for patients with inflammatory breast carcinoma.**

| Patient characteristics | Number (%) | Kaplan–Meier | |
|---|---|---|---|
| | | 5-OS (%) | Median OS (months) |
| Age | | | |
| <50 | 234 (26.5) | 52.7 | * |
| 50–59 | 262 (29.7) | 52.8 | 63 |
| 60–69 | 199 (22.5) | 45.4 | 40 |
| ≥70 | 188 (21.3) | 35.2 | 26 |
| Race | | | |
| White | 693 (78.5) | 49.4 | 58 |
| Black | 123 (13.9) | 29.7 | 25 |
| Unknown | 67 (7.6) | 54.9 | 61 |
| N stage | | | |
| N0 | 144 (16.3) | 61.6 | * |
| N1 | 372 (42.1) | 48.7 | 56 |
| N2 | 138 (15.6) | 48.5 | 49 |
| N3 | 177 (20.0) | 43.3 | 36 |
| $N_X$ | 52 (5.9) | 19.7 | 18 |
| M stage | | | |
| M0 | 572 (64.8) | 60.4 | * |
| M1 | 311 (35.2) | 24.0 | 21 |
| TNM stage | | | |
| III | 572 (64.8) | 60.4 | * |
| IV | 311 (35.2) | 24.0 | 21 |
| Surgery | | | |
| No surgery | 355 (40.2) | 29.3 | 21 |
| Not MRM | 162 (18.3) | 54.2 | * |
| MRM | 352 (39.9) | 60.5 | * |
| Unknown | 14 (1.6) | 32.5 | 29 |
| Radiation | | | |
| No | 487 (55.2) | 41.2 | 34 |
| Yes | 396 (44.8) | 54.2 | * |
| Chemotherapy | | | |
| No | 162 (18.3) | 25.1 | 15 |
| Yes | 721 (81.7) | 52.1 | 63 |
| Tumor size | | | |
| <2 cm | 68 (7.7) | 53.3 | * |
| 2–5 cm | 186 (21.1) | 54.0 | * |
| >5 cm | 257 (29.1) | 43.2 | 37 |
| Diffuse | 179 (20.3) | 40.3 | 29 |
| Unknown | 193 (21.9) | 50.1 | 66 |

**Table 3** (*continued*)

| Patient characteristics | Number (%) | Kaplan–Meier | |
|---|---|---|---|
| | | 5-OS (%) | Median OS (months) |
| Breast Subtype | | | |
| Her2-/HR+ | 293 (33.2) | 47.0 | 56 |
| Triple Negative | 200 (22.7) | 28.7 | 20 |
| Her2+/HR- | 151 (17.1) | 52.0 | 63 |
| Her2+/HR+ | 159 (18.0) | 71.3 | * |
| Unknown | 80 (9.1) | 39.6 | 22 |
| ER | | | |
| Negative | 386 (43.7) | 39.5 | 29 |
| Positive | 455 (51.5) | 54.7 | * |
| Unknown | 42 (4.8) | 35.8 | 21 |
| PR | | | |
| Negative | 499 (56.5) | 43.5 | 35 |
| Positive | 331 (37.5) | 54.2 | 66 |
| Other* | 53 (6.0) | 38.8 | 22 |
| Her-2 | | | |
| Negative | 496 (56.2) | 39.3 | 35 |
| Positive | 311 (35.2) | 62.7 | * |
| Other* | 76 (8.6) | 40.2 | 26 |
| Number of LN | | | |
| Negative | 104 (11.8) | 73.0 | * |
| Positive | 422 (47.8) | 52.8 | 63 |
| Other | 357 (40.4) | 34.2 | 25 |
| Marital status | | | |
| Married | 400 (45.3) | 53.6 | 66 |
| Divorce or widow | 238 (27) | 41.2 | 36 |
| Single | 201 (22.8) | 45.3 | 39 |
| Unknown | 44 (5.0) | 31.9 | 25 |

**Notes.**
*There are no results.

significantly improved 5-year OS by 15% (*Muzaffar et al., 2018*). However, RT was not an independent prognostic factor of CSS in IBC patients. Researchers have now realized that breast cancer is a systemic disease. Locoregional therapy is not enough to prevent distant metastasis. Therefore, systemic CT has been combined with surgery, RT, hormonal therapy and molecular targeted therapy. In this study, IBC patients treated with CT vs. without CT achieved a 5-year OS of 52.1% vs. 25.1% and a median OS of 63.0 vs. 15.0 months, respectively. These results are consistent with previous reports which showed that CT improved the 5-year OS of patients with IBC to 30–40% (*Fields et al., 1989*). Interestingly, our stratified analysis of surgical modalities based on TNM stage revealed that stage IV patients achieved a significant difference in survival when treated with MRM compared with non-MRM. Nevertheless, no difference was observed in stage III IBC patients. In this study, 118 (72.8%) patients in the non-MRM group underwent total
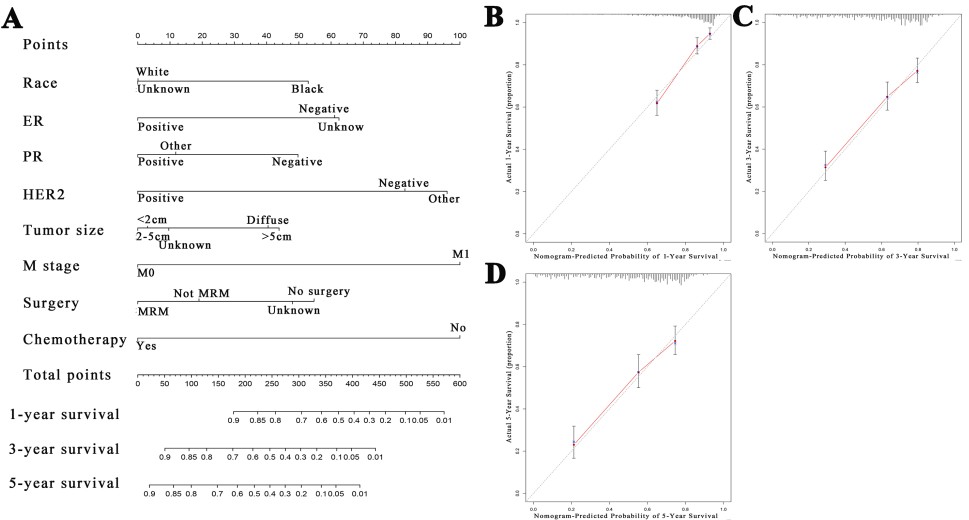

**Figure 1** **Nomogram and calibration curves of patients with inflammatory breast carcinoma.** (A) A nomogram for the prediction of 1-, 3- and 5-year cancer-specific survival in IBC patients. Calibration curves of the nomogram prediction of (B) 1-year, (C) 3-year, and (D) 5-year survival of patients with IBC.

(simple) mastectomy. Only nine (1.0%) patients underwent radical mastectomy and five (0.9%) patients underwent extended radical mastectomy. MRM is the most commonly used surgical approach of choice, as it allows both removal of the main tumor mass and adjacent glandular tissue with suspected infiltration and multifocality, and a sentinel axillary lymph node. This procedure significantly reduces primary tumor burden and peripheral subclinical lesions. Theoretically, it should bring marked benefits to both stage III and IV patients. A comprehensive explanation to rationalize these observations is still required. Lymph node status is also an important prognostic factor in IBC (*Liang et al., 2015*) and positive node status was reported to be a negative prognostic factor (*Wecsler et al., 2015*). However, N stage in our study was not statistically significant in multivariate analysis. A total of 355 patients in our study received no surgery; the N stage in these patients was categorized only by clinical features without pathology confirmation. We assume that inaccurate N stage or less important than M stage in multivariate Cox regression analysis could be the reasons why lymph node status was not included as a prognostic factor of IBC.

Trimodality treatment is recommended following the sequence of CT (including trastuzumab and hormonal therapy when necessary), then surgery and RT (*Robertson et al., 2010*). Two prospective randomized trials involving 68 patients with IBC treated with three cycles of CAF (cyclophosphamide, doxorubicin, and 5-fluorouracil) or CEF (cyclophosphamide, epirubicin, and 5-fluorouracil) followed by surgery, adjuvant therapy, and RT, reported a 5-year OS of 44% and a 10-year OS of 32% (*Baldini et al., 2004*). Our subgroup analysis based on treatment group including CT alone, CT combined with surgery, and CT combined with surgery and RT showed the advantage of trimodality treatment on survival. It was reported that patients with ER-negative IBC benefitted

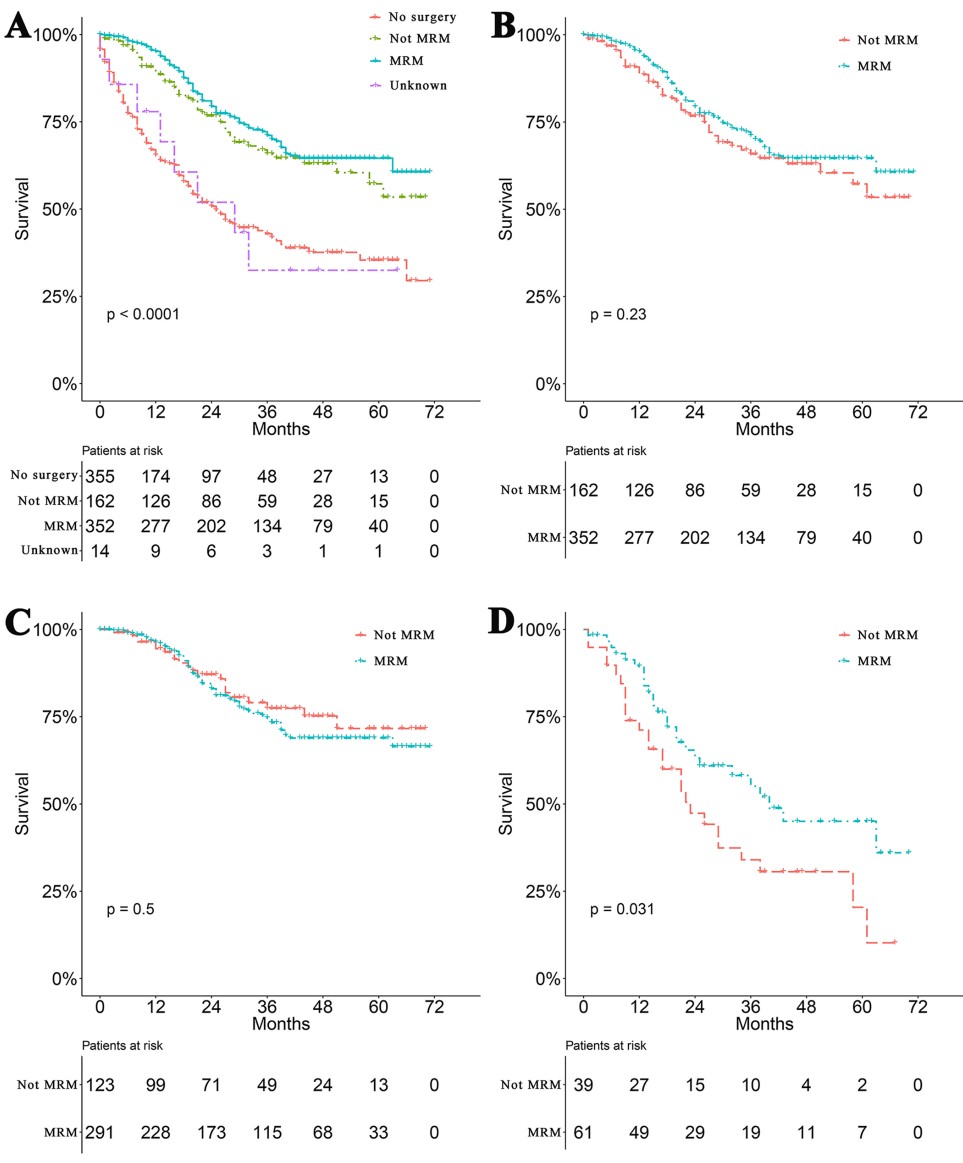

**Figure 2  Survival analyses based on surgery and stratified analyses TNM stage.** (A) Survival analyses of patients based on surgical modalities. (B) Survival analyses of patients with modified radical mastectomy (MRM) or non-MRM. Survival analyses of patients with MRM or non-MRM stratified by TNM stage. Five-year survival (C) stage III (MRM vs. non-MRM, 68% vs. 64.3%). (D) stage IV (MRM vs. non-MRM, 43.2% vs.18.9%).

from paclitaxel together with anthracycline-based regimens and obtained improved progression-free survival and OS (*Cristofanilli et al., 2007*; *Cristofanilli et al., 2004*). The optimal chemotherapeutic regimens and optimal combination of targeted treatments deserve further investigation.

To our knowledge, no nomogram model has been established to predict the survival of patients with IBC. As 34.4% of deaths are not attributed to primary breast cancer, CSS was used here to exclude the potential impact of other factors. It is worth mentioning that

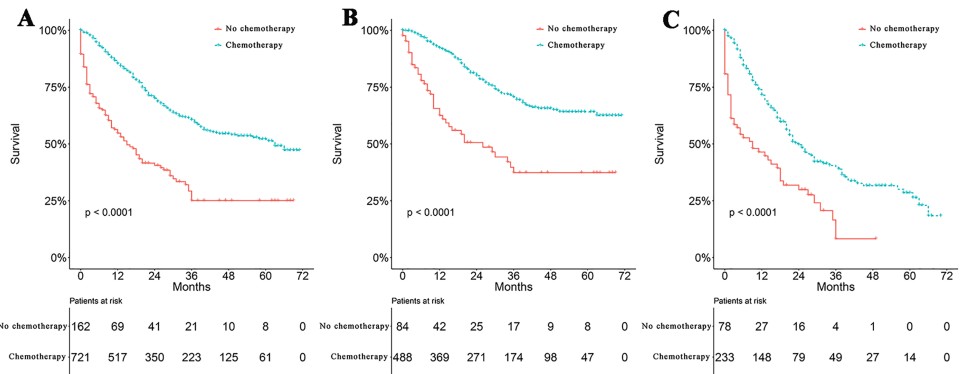

**Figure 3** **Survival analyses based on chemotherapy and stratified analyses based on TNM stage.** (A) Survival analyses of patients with IBC treated with or without chemotherapy (CT). Survival analyses of patients with or without CT stratified by TNM stage. (B) stage III (Yes vs. No, 70.7% vs. 37.4%) (C) stage IV (Yes vs. No, 39.7% vs. 8.3%).

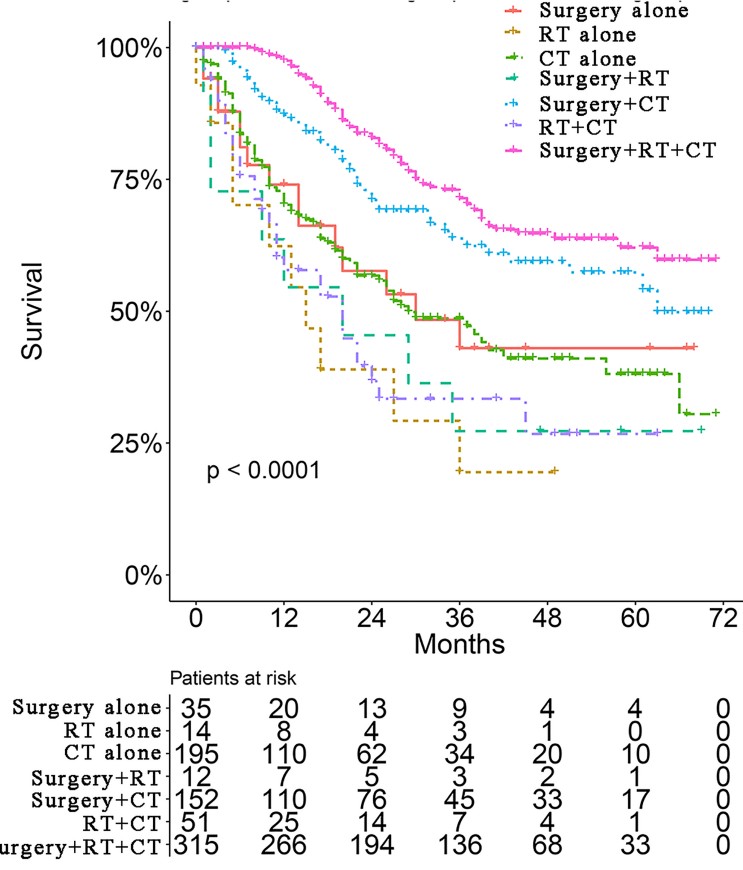

**Figure 4** **Survival analyses of patients with IBC treated with different strategies.**

our study included a large sample size of 883 patients which is difficult to achieve in a single institute clinical trial. In addition to race, Her-2 and hormone receptor status have been reported as prognostic factors for OS in IBC. All prognostic factors in the present study have not been reported in association with CSS. This nomogram included surgical modalities (MRM vs. non-MRM), and MRM was confirmed to be a protective factor for CSS compared to non-MRM. The nomogram with predictive accuracy was constructed based on comprehensive clinical pathological features in this study. However, there are some limitations that should be noted. The SEER database did not provide sufficient information on systemic therapy, such as CT regimens in patients with IBC, Her2-positive patients treated with or without trastuzumab, and hormone receptor-positive patients treated with or without hormonal therapy. Furthermore, limited information on local therapy, such as the sequence of RT and surgery, radiation dose and radiation fields, constrained our identification of more prognostic factors associated with detailed treatment information. Additionally, a prospective study should be conducted to validate the reliability of these results.

## CONCLUSIONS

A nomogram can be effectively applied to predict the 1-, 3- and 5-year survival of IBC patients. Our nomogram showed relatively good accuracy with a C-index of 0.735 and is a visualized individually predictive tool for prognosis. Treatment strategy greatly affected the survival of patients. Trimodality therapy was the preferable therapeutic strategy for IBC. Further prospective studies are needed to validate these findings.

## ACKNOWLEDGEMENTS

We thank Dr. Huang Yiwei for his partial statistical analysis in this study.

### Funding

This work was supported by Luoyang City Science and Technology Plan Medical and Health Project (Grant No. 1721001A-2).

### Grant Disclosures

The following grant information was disclosed by the authors:
Luoyang City Science and Technology Plan Medical and Health Project: 1721001A-2.

### Competing Interests

The authors declare there are no competing interests.

### Author Contributions

- Haige Zhang conceived and designed the experiments, performed the experiments, analyzed the data, prepared figures and/or tables, authored or reviewed drafts of the paper, approved the final draft.

- Guifen Ma performed the experiments, authored or reviewed drafts of the paper, approved the final draft.
- Shisuo Du analyzed the data, authored or reviewed drafts of the paper, approved the final draft.
- Jing Sun and Qian Zhang analyzed the data.
- Baoying Yuan contributed reagents/materials/analysis tools.
- Xiaoyong Luo authored or reviewed drafts of the paper, approved the final draft.

### Data Availability

The raw data is available as a Supplemental File. The raw data shows survival and prognostic factors of patients with inflammatory breast carcinoma.

### Supplemental Information

Supplemental information for this article can be found online at http://dx.doi.org/10.7717/peerj.7659#supplemental-information.

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
