# Peer review of "Nomogram for predicting cancer specific survival in inflammatory breast carcinoma: a SEER population-based study"

_PeerJ, doi:10.7717/peerj.7659_

## Round 0.1 · original submission · Major Revisions

Dear authors,

In light of the reviewers' comments, I think your manuscript has high scientific standards to be considered for publication in PeerJ. However, there are some modifications which you must apply in a revised version of the text (MAJOR REVISION). Please, see the comments below so as to have more information.

With respect and kind regards,
Dr Palazón-Bru (academic editor for PeerJ)

Reviewer 1 ·

Basic reporting

No comment

Experimental design

No comment

Validity of the findings

No comment

Additional comments

- There is a lot of time series, why did the authors select data from 2010 to 2015?
- The authors have to compare their performance with the previous works on the same dataset.
- Quality of figures needs to be improved. Now it is hard to see.
- The authors reported C-index values. It is good, however, there is necessary for reporting the other metrics, such as an important one is the ROC Curve.
- In their tables (1 vs 2), what is the "1[Reference]"?
- Why there are missing p-values in the multivariate analysis?
- It is important to discuss the characteristics with high p-value (not significant)
- It is necessary to test the prediction model with an independent dataset.
- There are some grammatical errors and typos. The authors should re-check and revise carefully.

Reviewer 2 ·

Basic reporting

Clear.
Background not sufficient, should have more introduction about similar studies using SEER database.

Experimental design

Well done.

Validity of the findings

The majority of the findings is valid.

Additional comments

The authors used the SEER database and analyzed the prognostic factors of inflammatory breast cancer patients. They used univariate and multivariate analysis to screen for significant prognostic factors and develop a survival nomogram that might help to provide prognostic information of IBC patients. I have several comments.


1, The authors stated in the discussion that “To our knowledge, this is the first study to analyze the survival of a large 163 cohort of patients with IBC based on the SEER database and” This is definitely not true. There are numerous published papers using the SEER database to analyzed the prognosis and prognostic factors of IBC patients.

2, The generation of the nomogram has a potential mistake. All of the variables, e.g. ER, PR, Tumor size, etc, has the same increase of points when different values were selected. (Namely, for ER, positive to unknown is equal to negative to positive). This is not correct as the HR between pos and unknown should not be 1/2 of the HR between negative and unknown. To solve this problem , the authors could set the variables as factor in R, using the as.factor codes.

3, the comparison between MRM and non-MRM is confusing. What do you mean by non-MRM? Do you mean breast conserving surgery? BCS is almost contraindicated in IBC patients. Do you mean radical mastectomy or extended mastectomy? They removed more tissues than MRM, why inferior to MRM?

4, I think the reason that LN is not significant is possibly due to the co-existence of M stage in the multivariate analysis. The authors could check this.

5, there are many others studies using SEER database to investigate the IBC patients. The authors should not avoid them, and should present the novelty of the current studies, when compared with previous studies.

---

## Round 0.2 · accepted · Accept

Dear authors,

I am pleased to inform that your paper has been accepted for publication in PeerJ.

Congratulations!

With respect and kind regards,
Dr Palazón-Bru (academic editor for PeerJ)

# Reviewer 1 ·

Basic reporting

No comment

Experimental design

No comment

Validity of the findings

No comment

Additional comments

My previous comments have been addressed.